# Characterization of Meteorological Droughts Occurrences in Côte d'Ivoire: Case of the Sassandra Watershed

**Natacha Santé [1], Yao Alexis N'Go [1], Gneneyougo Emile Soro [1], N'Diaye Hermann Meledje [2] and Bi Tié Albert Goula [1,\*]**

[1]   Unit Training and Research in Science and Environment Management, University Nangui Abrogoua, 02 BP 801 Abidjan 02, Côte d'Ivoire; natachasante@gmail.com (N.S.); nyaoalexis@yahoo.fr (Y.A.N.G.); ge_soro@yahoo.fr (G.E.S.)

[2]   Ecology Research Center, Marine Geology Laboratory, Sedimentology and Environment, 08 BP 109 Abidjan 08, Côte d'Ivoire; meledjendiay@yahoo.fr

\*   Correspondence: goulaba2002@yahoo.fr; Tel.: +225-0710-4569

**Abstract:** The Sassandra Basin, like most regions of Côte d'Ivoire, is increasingly affected by droughts that involve many environmental, social and economic impacts. This basin is full of several amenities such as hydroelectric dams, hydraulic and agricultural dams. There is also a strong agricultural activity. In the context of climate change, it is essential to analyze the occurrence of droughts in order to propose mitigation or adaptation measures for water management. The methodological approach consisted initially in characterizing the dry sequences by the use of the SPI (Standardized Precipitation Index) and secondly in determining the probabilities of occurrence of successive dry years using by Markov chains 1 and 2. The results indicate that most remarkable droughts in terms of intensity and duration occurred after the 1970s. A comparison of Markov matrices 1 and 2 between the period considered 1953–2015 with the periods 1953–1970 and 1971–2015 shows a profound change in the distribution of droughts at the different station. Thus, the probability of having two successive dry years is greater over the period 1970–2015 and is accentuated to the Southern and Northern regions (probabilities ranging from 71% to 80%) of the basin. Over the 1970–2015 period, the probability of obtaining three successive dry years is significantly high in this watershed (between 20% and 70%).

**Keywords:** drought; sassandra watershed; Côte d'Ivoire

## 1. Introduction

Drought is one of the greatest natural hazards with effects on water resources, natural ecosystems and agriculture. Frequent and severe droughts limit the development of vegetation cover and make the soil more susceptible to erosion by leaching due to heavy rainfall [1]. They are responsible for famine, epidemics and land degradation in developing countries and cause major economic losses in developed regions [2]. From a meteorological point of view, drought can be defined as an abnormal but recurrent behavior of the climate essentially linked to the absence of rainfall received by a region within a certain period of time [3,4].

West and Southern Africa are experiencing severe drought, disrupting agricultural and livestock production systems in nearly 14 countries. Agriculture is nearly 95% rainfed in the region. It therefore remains highly vulnerable to rainfall fluctuations [5]. Work on climate fluctuations in this part of the world has made it possible to identify periods of drought since the 1970s. In Côte d'Ivoire, this deficit situation has resulted in major climatic disruptions, including a significant drop in rainfall [6–10]. There has also been an abnormal extension of the dry season [11], irregular and uneven rainfall

distribution and a significant decline in hydroelectric production [12]. These disruptions have had serious consequences such as forest and plantation fires accompanied by a sharp drop in agricultural production and power outages. For example, in December 1983, fires destroyed 60,000 ha of forests and 108,000 ha of plantations and crops [13]). Given the magnitude of the environmental impacts of droughts, public authorities should attach greater importance to the development of an early warning and adaptation strategy that would announce the beginning, end and future intensity of drought. The example of the Sassandra Watershed chosen for this study is interesting because this region undergoes more and more dry season sequences. These deficit periods caused a disruption of cropping seasons in rural areas [14,15] and the decrease in in stream flows [16,17]. This basin is mainly marked by strong anthropogenic pressures. Indeed, this basin, which is also part of Côte d'Ivoire's cocoa and coffee economy, is experiencing a reduction in plant cover linked to systematic large-scale deforestation of the forest heritage for the creation of plantations [14]. There are socio-economic infrastructures (hydroelectric dams, agricultural dams, etc.) and this basin are subject to many water-related projects.

Given the impact and occurrence of droughts that are likely to increase in the coming years under certain scenarios of global change [18], it is essential to better understand how irregularity and rainfall distribution is manifested and to adopt preventive measures [19]. It is in this context that the present study was initiated on the Sassandra watershed. This study aims to highlight the occurrence of meteorological droughts in this basin using the Markov chains method based on annual rainfall over the period 1953–2015.

## 2. Study Area

The Sassandra basin is located between longitude 5°75 and 8°16 West and latitude 5° and 9°75′ North (Figure 1). It covers Odienné, Touba, Seguéla, Daloa, Man, Guiglo, Soubré, Sassandra, Gagnoa cities. It has a total area of about 75,000 km$^2$, of which the Ivorian part occupies an area of about 67,000 km$^2$. The relief of the study area consists of plains and uplands at varying altitude from 1100 to 1180 m. There are some rock chains that have resisted to erosion. The zenith sun movement controls the migration of the ITCZ (Intertropical Convergence Zone) in Côte d'Ivoire, which explains the introduction of different seasonal regimes. Thus the basin of Sassandra is subdivided into four climatic units according to rainfall patterns [20]. The equatorial transitional climate with four seasons (a large rainy season from April to June, a small rainy season from September to November, a large dry season from December to March and a small dry season from July to August). The interannual rainfall average is 1441.5 mm; the equatorial climate of attenuated transition is marked by two seasons (a major rainy season covering the months of August to October and a major dry season from November to March). The interannual average is 1305.2 mm; the tropical transitional climate has a unimodal pattern. It is characterized by a rainy season that occurs from June to October. The dry season covers the months of November to March. The interannual rainfall recorded at the Odienné station is 1473 mm; the mountain climate is characterized by an azonal type pattern. The highest rainfall peak is recorded in September (279 mm). The dry season covers the month of November to March. The average interannual rainfall is 1578.5 mm. The average monthly temperatures range from 23°C to 28°C and are generally uniform from one region to another. The average monthly relative humidity varies from 77 to 96% in Guinea environment and from 44 to 83% in the North [21].

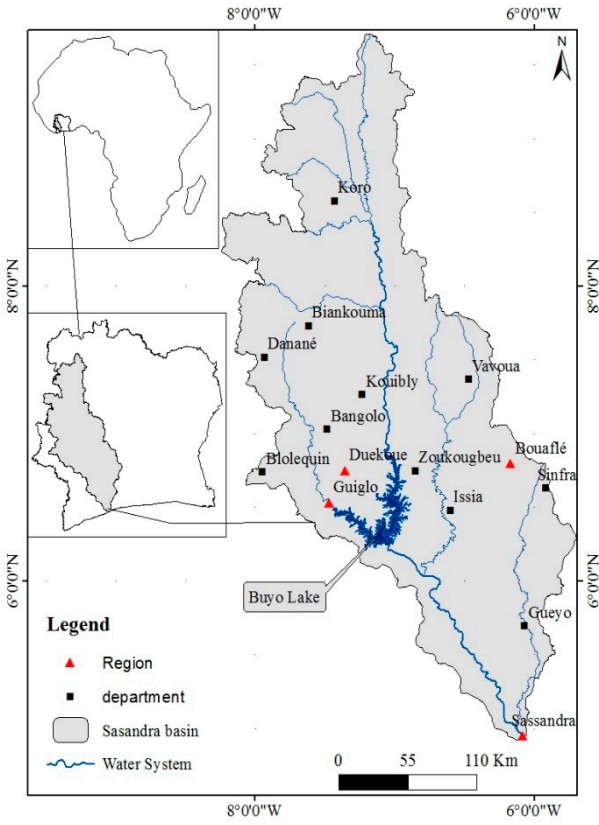

**Figure 1.** Geographic localization of the study area.

## 3. Data and Methods

### 3.1. Historical Time Series Data

Historical rainfall data used in this work cover the entire study area at the yearly scale. Climate data over the period 1953–2015 was provided by the National Meteorology Directorate (DMN) of the Society of Development and Exploitation, Aeoroportuary, Aeronautic and Meteorology (SODEXAM) of Côte d'Ivoire. These data consist of daily rainfall readings from ten rainfall stations (Table 1) selected to provide the most homogeneous coverage of the different climatic areas across in the Sassandra basin. The study variable is the annual rainfall for the period 1953–2015.

**Table 1.** Rainfall stations selected for the study.

| Climatic Area | Name | Latitude North | Longitude West | Code |
|---|---|---|---|---|
| Attiean climate | Sassandra | 4°57′ | 6°50′ | 1090017200 |
| | Gagnoa | 6°07′ | 5°56′ | 1090010300 |
| Baoulean climate | Soubré | 5°47′ | 6°36′ | 1090018100 |
| | Guiglo | 6°32′ | 7°28′ | 1090011200 |
| | Vavoua | 7°22′ | 6°28′ | 1090021400 |
| | Seguéla | 7°57′ | 6°40′ | 1090017500 |
| | Daloa | 6°53′ | 6°27″ | 1090008200 |
| | Touba | 8°17′ | 7°41′ | 1090020500 |
| Mountain climate | Man | 7°24′ | 7°31′ | 1090014200 |
| Sudanese climate | Odienné | 9°30′ | 7°34′ | 1090016000 |

*3.2. Methodology*

3.2.1. Characterization of Meteorological Drought Sequences

❖ *The choice of the statistical index*

The Standardized Precipitation Index (SPI), developed by [22], is used in this study to characterize meteorological droughts. It has advantages in terms of statistical consistency and the ability to describe both short-term and long-term drought impacts through different time scales [22]. The development of this index is based solely on the use of rainfall as a baseline data to determine wet and dry periods, and to specify their duration and intensity. The probabilistic nature of the SPI index allows it to be comparable between different sites [23].

❖ *Standardized Precipitation Index*

The standardized precipitation index (SPI) [22,24] was developed to quantify the rainfall deficit for multiple time scales that will reflect the impact of drought on the availability of different types of water resources over a given period. It is expressed as follows (Equation (1)):

$$SPI = (Pi - Pm)/S \qquad (1)$$

Pi: Total rainfall over year i (mm); Pm: Average precipitation over the period 1953–2015 (mm); S: Standard deviation of precipitation over the period 1953–2015 (mm).

According to [22], a drought occurs when the SPI is consecutively negative and its value reaches an intensity of −1 or less and ends when the SPI becomes positive. A drought classification is performed according to the SPI values (Table 2).

**Table 2.** Classification of drought sequences according to Standardized Precipitation Index (SPI) [22].

| SPI Value | Drought Sequence |
|---|---|
| −0.99 to 0.99 | Near the Normal |
| −1.00 to –1.49 | Moderately dry |
| −1.50 to –1.99 | Severely dry |
| −2.00 and under | Extremely dry |

❖ *Descriptive parameters of drought sequence*

- *Maximum duration of drought sequences*

Duration is an important characteristic of drought. In fact, if a drought starts quickly under some weather conditions, it usually takes at least two to three months before it can spread to other regions. It can then persist for months or even years. The calculation of the duration is as follows [25]. (Equation (2)).

$$D = (A_{end} - A_{initial}) \qquad (2)$$

$A_{initial}$: Year of the initial dry period; $A_{end}$: Year of end of the dry period

- *Intensity of drought sequences*

Intensity of drought can be defined as the magnitude and severity of the consequences for the rainfall deficit. It can be evaluated using the SPI values. In this study, the extreme value of the SPI was considered as a reference value for drought intensity.

3.2.2. Time Series Change Detection

Rupture is defined as a sudden change in the properties of a random process [26]. Rupture tests are complementary to standard indices because the existence of sudden change in time series is a

possible cause of the rupture of the homogeneity of these series [27]. In this study, the Cumulative Gap (CG) test was used to detect possible sudden changes in rainfall series. This non-parametric procedure, based on rank, analyzes whether the means of the two parts of the series are different for an unknown break date [28]. The statistic of this test is calculated from the cumulative sum of the "sign" function of the difference between the observed values and the median. This statistical processing is performed with Hydrospect 2.0 software. The test statistic is defined as follows (Equation (3)):

$$|TS| = (2 \backslash n) \max |S_k| \text{ with } S_k = \sum_{i=1}^{k} \text{sign} (x_i - X_m) \text{ and } (k = 1, \ldots, n) \tag{3}$$

where $x_i$ is the extreme hydrometric observation of rank $i$ ($i = 1 \ldots n$); $X_m$ is the median of the extreme hydrometric series; $S_k$ is the statistic test; $n$ is the number of value for the rank $i$.

### 3.2.3. Characterization of Meteorological Droughts Occurrence by Markov Chains

Several statistical techniques for analyzing precipitation data have been published in the literature. The most used technique is still the one based on the Markov chains. This method is widely used for rainfall analysis and modelling [29–36]. A Markov string is a series of random variables ($Xn, n \in N$) that allows to model the dynamic evolution of a random system: $Xn$ represents the state of the system at time $n$. The fundamental property of Markov chains, known as "Markov property", is that its future evolution depends on the past only through its current value. In other words, conditionally to $Xn$ ($X0$, $\ldots$, $Xn$) and ($Xn+k, k \in N$) are independent [37].

○  *Markov chain with two states of order 1*

For a first order Markov chain, the state of the variable $E(t)$ at time $t$ depends only on its state at time ($t − 1$). Thus, we have four situations: [31]

$$\begin{aligned}
P_{00} &= pr(E(t + 1) = 0| (E(t) = 0)) \\
P_{01} &= pr(E(t + 1) = 1 | (E(t) = 0)) \\
P_{10} &= pr(E(t + 1) = 0 | (E(t) = 1)) \\
P_{11} &= pr(E(t + 1) = 1| (E(t) = 1))
\end{aligned} \tag{4}$$

$P_{ij}$ is the probability of going to state $j$ knowing that you are in state $i$. These probabilities were calculated using the following relationship:

$$P_{ij} = N_{ij}/N_i \text{ with: i and j} = 0 \text{ or } 1 \tag{5}$$

$N_{ij}$ is the transition number from state $i$ to state $j$ and $N_i$ is the number of transitions from state $i$ to any other state. The pairs of years $N_{ij}$ are determined [35] (Equation (6)):

$$\begin{cases}
N_0 = N_{00} + N_{01} \\
N_1 = N_{10} + N_{11} \\
N = N_0 + N_1
\end{cases} \tag{6}$$

$N_0$; $N_1$ and N are the number of dry, wet years and the total number of years of observation, respectively. $N_{01}$ and $N_{10}$ respectively represent the number of years of state change from a dry year to a wet year and from a wet year to a dry year. The transition matrix P of the conditional probabilities $P_{ij}$, is presented so that each line is equal to 1 [35]. Resulting in a set of possible $P_{ij}$ values (Equation (7)):

$$P = \begin{bmatrix}
P_{00} & P_{01} & \ldots \\
P_{10} & P_{11} & \ldots \\
\ldots & \ldots & \ldots \\
P_{i0} & P_{i1} & \ldots
\end{bmatrix} \tag{7}$$

○    *Markov chain with two states of order 2*

For a Markov string of order 2, the state of the variable E(t) at time t depends on its state E(t − 1) at time (t − 1) as well as its state E(t − 2). The probability of having this state can be written:

$$P_{ijk} = pr\ (E(t) = k\ |(E(t − 1) = j, E(t − 2) = i)) \tag{8}$$

$P_{ijk}$ represents the conditional probability of having a state doublet (*j, k*) following the state doublet (*i, j*) and *i, j, k* = 0 or 1, calculated using the following relationship [31]:

$$P_{ijk} = N_{ijk}/N_{ij} \tag{9}$$

where $N_{ijk}$ is the number of transitions from the state doublet (i, j) to the state doublet (j, k).

The process of transition of conditional probabilities with the Markov 2 chain is as follows (Table 3):

**Table 3.** Markov process of order 2 [35].

| State at Day k−1 and k−2 | State at Day k−1 and k | | | |
|---|---|---|---|---|
| | **00** | **01** | **10** | **11** |
| **00** | P000 | P001 | 0 | 0 |
| **01** | 0 | 0 | P010 | P011 |
| **10** | P100 | P101 | 0 | 0 |
| **11** | 0 | 0 | P110 | P111 |

## 4. Results

### 4.1. Analysis of Meteorological Drought Sequences

The application of the cumulative gap test identified change point detection in the series. The null hypothesis of no rupture was rejected at the 99% and 95% confidence levels. These ruptures were identified mainly after 1970 except in the Man, Guiglo and Gagnoa regions, which had their ruptures between 1967 and 1969. Temporal analysis shows a slight downward trend in SPIs after ruptures, confirming a decrease in rainfall. SPI values over the 1953–2015 period show very few dry sequences before the rupture years (Figure 2).

For the equatorial transition regime (attiean climate), change point detection were detected in 1966 and 1983 at the Gagnoa and Sassandra stations respectively. Before these change point detection, the index counted four dry sequences in Gagnoa and three dry sequences in Sassandra. Thus, the most remarkable sequences have a duration of nine successive years in Gagnoa and 18 in Sassandra.

For the equatorial regime of attenuated transition (Baoulean climate): before the change point detection years, the SPI index recorded four dry sequences in Soubré, three in Guiglo, seven in Daloa, four in Vavoua, 12 in Seguéla and seven in Touba. In this period, the Daloa and Touba stations recorded two sequences of two successive dry years. For Daloa, the index shows the periods 1964–1965 and 1969–1970 and for Touba, the periods 1960–1961 and 1970–1971. As for the Seguela station, the index detected a sequence of five successive dry years. The most remarkable dry episodes after the rupture have lasted 13 successive years in Soubré, six years in Guiglo, seven years in Daloa, four years in Vavoua, eight years in Seguéla and 10 years in Touba.

For the mountain regime, the Man station recorded five dry sequences before 1968. This number increased to 27 after the change point detection. The index has also recorded sequences of successive dry years, the most remarkable of which is five years (2011–2015).

For the Tropical Attenuated Transition Regime (Sudanese climate), Odienné station recorded 8 dry sequences, including a sequence of two and three successive dry years in 1980–1981 and 1973–1975. After the change point detection, the SPIs show 24 dry sequences, the most remarkable in terms of duration being 11 successive dry years (1983–1993).

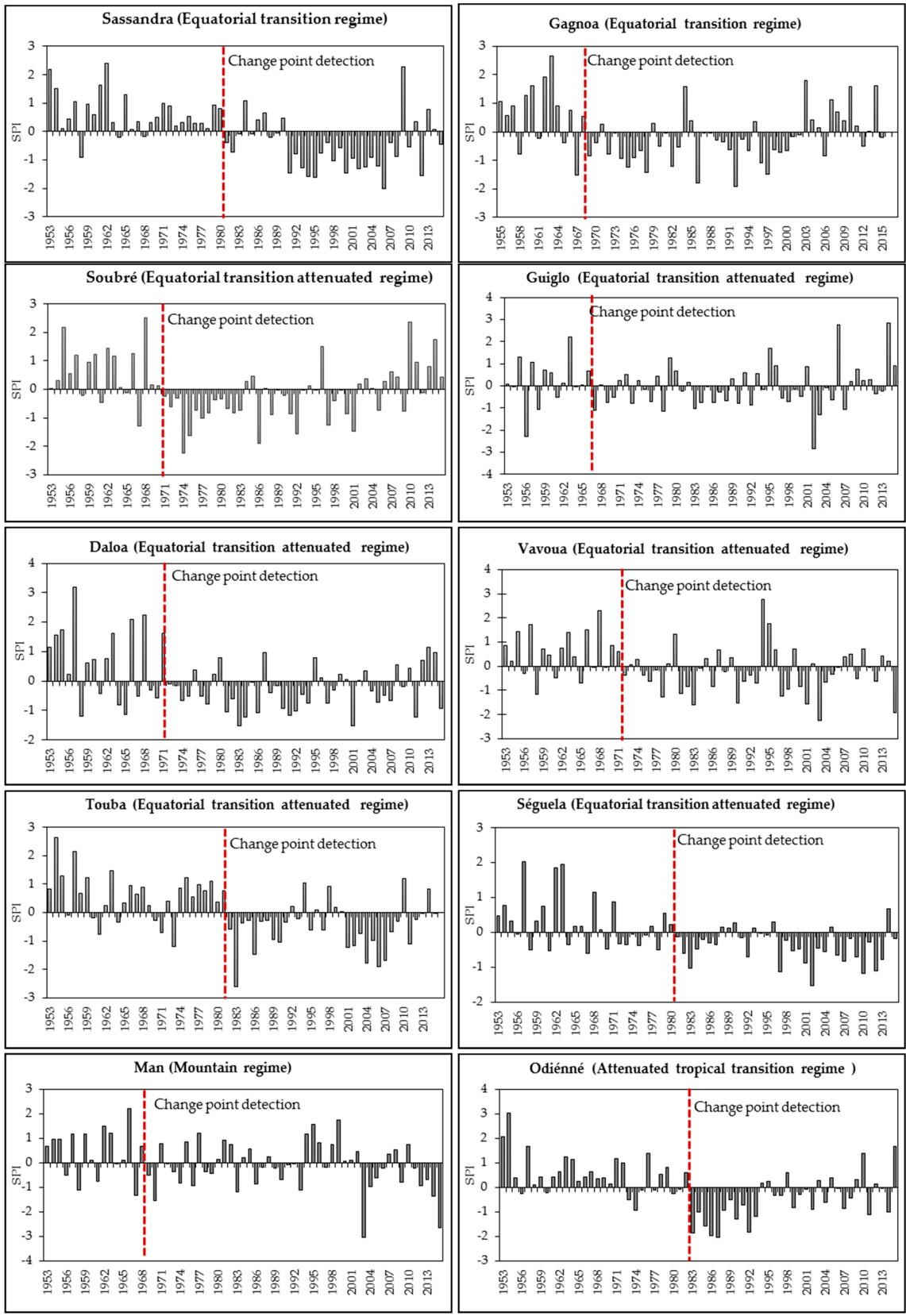

**Figure 2.** SPI index evolution over the period 1953–2015.

### 4.2. Intensity and Duration Parameter Analysis

The most remarkable drought years in terms of intensity and duration during the 62 years analyzed differ according to the climatic zones of the watershed (Table 4). For the attean climate, the driest years were those of 1992 and 1998. These dry events were classified as "very dry" in terms of intensity. The Gagnoa station recorded the highest intensity. As for the Sassandra station, it experienced the longest drought period with 18 years of consecutive dry sequences.

At the level of the Attenuated Equatorial Transition Regime (Baoulean climate), the stations of Soubré, Guiglo, Vavoua and Touba recorded the highest intensities in terms of drought. These events characterized as "extremely dry" were detected in 1974, 2002, 2003 and 1983, respectively. In this region of the basin, the Guiglo station was the most affected in terms of intensity while the Touba station recorded the longest dry period (10 successive dry years).

In the Tropical transitional Attenuated Regime (Sudanese climate), the most remarkable droughts were those of 1987. These dry episodes were described as extremely severe in terms of intensity. Odienné station was marked by a longer dry period with (11 successive years).

As for the Mountain Regime, it recorded a deficit of an intensity described as extremely severe in 2003. This area was marked by a long drought period of seven years.

The application of the SPI to rainfall data for the period 1953–2015 shows the western part of the basin, which is the most affected by drought in terms of intensity (−3.06) while the southern part of the basin has the longest drought period (18 years).

**Table 4.** Intensity and duration of meteorological drought sequences during the period 1953–2015.

| Stations | Intensity (SPI) | Type | Maximum Duration | Date of Occurrence |
|---|---|---|---|---|
| Daloa | −1.53 | Severely dry | 7 | 2001 |
| Vavoua | −2.26 | Extremely dry | 4 | 2003 |
| Man | −3.06 | Extremely dry | 5 | 2003 |
| Guiglo | −2.86 | Extremely dry | 6 | 2002 |
| Seguéla | −1.53 | Severely dry | 8 | 2002 |
| Touba | −2.61 | Extremely dry | 10 | 1983 |
| Odienné | −2.05 | Extremely dry | 11 | 1987 |
| Gagnoa | −1.90 | Severely dry | 7 | 1992 |
| Soubré | −2.25 | Extremely dry | 13 | 1974 |
| Sassandra | −1.87 | Severely dry | 18 | 1998 |

### 4.3. Analysis of the Meteorological Droughts Occurrence

#### 4.3.1. Transition States Probability of Markov Chains 1

The probability of obtaining two successive dry years and two successive wet years is high in the Gagnoa and Sassandra areas (reaching 70%). In this climatic zone in the southern part of the basin, when a dry year is followed by a non-dry year or a non-dry year is followed by a dry year, the probability is low (Table 5). The Soubré, Seguéla, Daloa and Touba regions recorded high probabilities (over 50%) of obtaining a doublet of successive dry years. As for the Guiglo, Vavoua and Man regions, these probabilities were average. The probabilities of having a wet year after a dry year and a wet year after a wet year are low in these regions of the basin. The Man region recorded average probabilities (55%) for two successive dry years and two successive wet years. As for the other probabilities, they are very low in this climate regime.The chances of obtaining successively dry and successively wet episode doublets are very high in Odienné. When a year is dry and is followed by a humid year or a humid year is followed by a dry year, the probabilities are very low. The analysis of the occurrences of two successive dry years over the 1953–2015 series shows very high probabilities over the entire basin (62% on average). However, the regions of Sassandra, Gagnoa, Seguéla, Touba and Odienné recorded the highest probability (up to 70%) of dry spells over this period.

**Table 5.** Occurrence of meteorological droughts using Markov Chains 1 over the period 1953–2015.

| Climate Regimes | Stations | Probability (%) | | | |
|---|---|---|---|---|---|
| | | W-W | D-W | W-D | D-D |
| **Attiean climate** | Sassandra | 68 | 30 | 30 | 70 |
| | Gagnoa | 54 | 32 | 46 | 70 |
| Baoulean climate | Soubré | 57 | 36 | 40 | 64 |
| | Guiglo | 43 | 55 | 57 | 45 |
| | Vavoua | 40 | 52 | 60 | 50 |
| | Seguéla | 46 | 31 | 54 | 70 |
| | Daloa | 44 | 40 | 56 | 60 |
| | Touba | 62 | 30 | 38 | 70 |
| Mountain climate | Man | 55 | 45 | 45 | 55 |
| Soudanese climate | Odienné | 60 | 38 | 40 | 63 |

Note: D: Dry year; W: Humid year.

The results of the occurrence analysis for two successive dry years before and after 1970 are presented in Table 6.

**Table 6.** Occurrence of meteorological droughts using Markov Chains 1 over the periods 1953–1970 and 1971–2015.

| Period | Climate Regimes | Stations | Probability (%) | | | |
|---|---|---|---|---|---|---|
| | | | W-W | D-W | W-D | D-D |
| 1953−1970 | Attiean climate | Sassandra | 28 | 36 | 71 | 60 |
| | | Gagnoa | 38 | 50 | 63 | 50 |
| | Baoulean climate | Soubré | 25 | 70 | 75 | 30 |
| | | Guiglo | 38 | 60 | 63 | 40 |
| | | Vavoua | 11 | 78 | 67 | 22 |
| | | Seguéla | 29 | 40 | 71 | 55 |
| | | Daloa | 40 | 63 | 60 | 23 |
| | | Touba | 44 | 45 | 60 | 45 |
| | Mountain climate | Man | 40 | 56 | 55 | 44 |
| | Soudanese climate | Odienné | 50 | 15 | 50 | 85 |
| 1971−2015 | Attiean climate | Sassandra | 77 | 20 | 23 | 80 |
| | | Gagnoa | 55 | 39 | 45 | 61 |
| | Baoulean climate | Soubré | 61 | 36 | 40 | 64 |
| | | Guiglo | 35 | 54 | 65 | 46 |
| | | Vavoua | 41 | 50 | 59 | 45 |
| | | Seguéla | 61 | 22 | 39 | 78 |
| | | Daloa | 48 | 45 | 52 | 55 |
| | | Touba | 70 | 30 | 30 | 71 |
| | Mountain climate | Man | 58 | 43 | 42 | 55 |
| | Soudanese climate | Odienné | 55 | 39 | 41 | 61 |

Note: D: Dry year; W: Humid year.

During the period 1953–1970, the probability of obtaining two successive humid years (W-W) and two consecutive dry years (D-D) is relatively low in the basin (less than 50%) except at the Odienné, Sassandra and Seguéla stations which have a high probability (85%, 60% and 55%, respectively) of obtaining two successive dry years (D-D). When a year is dry, the probability of having the following year not dry (D-W) is high at the Daloa (70%), Soubré (70%) and Vavoua (78%) stations. In the event

that a year is not dry, the probability that the following year will be dry is high throughout the basin (greater than 50%) and varies from 55% to 75%.

Over the period 1971–2015, a trend contrary to the previous period is observed. Indeed, if a year is dry at the beginning, the probability of having a dry year (D-D) is high for most regions (above 50%) and reaches its maximum (80%) at the Sassandra station. In the case where the dry year is followed by a non-dry year (D-W), the probability is relatively low and varies from 25% to 48% except in the Guiglo area (54%). When a humid year is followed by a dry year (W-D), the probability is high at Guiglo (62%) and Vavoua (59%). For the other parts of the basin, this probability is moderate. When two years are non-dry successively (W-W), the probability is greater than 50% over most of the basin except in the Daloa, Vavoua and Guiglo regions where the values are low (less than 50%).

The analysis shows the increase in drought episodes over the period 1971–2015. This is reflected in the high probabilities, the most notable of which were recorded in the North (Touba (71%) and Seguéla (78%)) and the South (Sassandra (80%)) of the basin.

4.3.2. Transition States Probability of Markov Chains 2

The results of the Markovian approach on the 1953–2015 series show that the probability of obtaining a dry year after two successive dry years (D-D-D) is high in the areas of Sassandra (67%) and Gagnoa (59%). As for the stations of Seguéla (51%), Soubré (55%) and Touba (50%) located in an equatorial regime of attenuated transition (Baoulean climate), they recorded average values. At the other stations, this probability is low (<50%) (Table 7). The probability of having a dry year between two wet years (W-D-W), a wet year followed by two successive dry years (W-D-D) and a wet year after two dry years (D-D-W) is very low over the entire basin. The averages recorded are 24%, 20.4% and 16.9%, respectively. Overall, the probability of having three consecutive dry years (D-D-D) is relatively low in the basin (44.8% on average). The most affected regions are those in the southern part of the basin.

**Table 7.** Occurrence of meteorological droughts using Markov Chains 2 over the period 1953–2015.

| Climate Regimes | Stations | Probability (%) | | | |
|---|---|---|---|---|---|
| | | W-D-W | W-D-D | D-D-W | D-D-D |
| Attiean climate | Sassandra | 15 | 6 | 7 | 67 |
| | Gagnoa | 23 | 20 | 11 | 59 |
| Baoulean climate | Soubré | 30 | 10 | 10 | 55 |
| | Guiglo | 40 | 17 | 20 | 30 |
| | Vavoua | 35 | 25 | 21 | 25 |
| | Seguéla | 25 | 25 | 15 | 51 |
| | Daloa | 22 | 32 | 22 | 40 |
| | Touba | 14 | 21 | 18 | 50 |
| Mountain climate | Man | 22 | 25 | 23 | 30 |
| Soudanese climate | Odienné | 16 | 23 | 22 | 41 |

Note: D: Dry year; W: Humid year.

The results of the occurrence of three successive dry years over the periods 1953–1970 and 1971–2015 are shown in Table 8.

During the period 1953–1970, the probability of having three successive dry years is relatively low at all stations except the Odienné station where this probability is 60%. In the regions of Man, Soubré, Vavoua and Daloa, the probability of having three successive dry years is zero.

During the period 1971–2015, the probability to observe (D-D-D) is high and higher than 50% at Seguéla (60%), Touba (63%) and the maximum is reached at Sassandra (70%). As for the rest of the stations, the probability is low and less than 50%.

**Table 8.** Occurrence of meteorological droughts using Markov Chains 2 over the periods 1953–1970 and 1971–2015.

| Period | Climate Regimes | Stations | Probability (%) | | | |
|---|---|---|---|---|---|---|
| | | | W-D-W | W-D-D | D-D-W | D-D-D |
| **1953−1970** | Attiean climate | Sassandra | 29 | 43 | 20 | 30 |
| | | Gagnoa | 38 | 25 | 10 | 20 |
| | Baoulean climate | Soubré | 50 | 25 | 20 | 0 |
| | | Guiglo | 50 | 13 | 20 | 20 |
| | | Vavoua | 60 | 22 | 25 | 0 |
| | | Seguéla | 14 | 57 | 30 | 20 |
| | | Daloa | 40 | 20 | 20 | 0 |
| | | Touba | 22 | 33 | 30 | 15 |
| | Mountain climate | Man | 33 | 33 | 30 | 0 |
| | Soudanese climate | Odienné | 0 | 60 | 20 | 60 |
| **1971−2015** | Attiean climate | Sassandra | 10 | 10 | 10 | 70 |
| | | Gagnoa | 25 | 20 | 17 | 43 |
| | Baoulean climate | Soubré | 20 | 20 | 20 | 50 |
| | | Guiglo | 40 | 30 | 21 | 25 |
| | | Vavoua | 30 | 30 | 30 | 20 |
| | | Seguéla | 11 | 25 | 15 | 60 |
| | | Daloa | 20 | 30 | 30 | 30 |
| | | Touba | 20 | 10 | 8 | 63 |
| | Mountain climate | Man | 17 | 25 | 0 | 25 |
| | Soudanese climate | Odienné | 20 | 23 | 22 | 40 |

Note: D: Dry year; W: Humid year.

### 4.4. Analysis of the Spatial Variability of Drought Occurrence

4.4.1. Spatial Variability of the Probability for Two Successive Dry Years

Figure 3 shows the probability spatial distribution for two successive dry years (D-D) over the Sassandra basin for the periods 1953–2015, 1953–1970 and 1971–2015. The period 1953–2015 is marked by an increase in meteorological droughts over almost the entire basin with the probability of having a very high doublet of dry years that reaches 70%. The most affected areas are those in the North and South of the catchment area.

From 1953 to 1970, the drought was particularly severe in the extreme north of the basin (around the Odienné region) and in the south (precisely around Sassandra) with probabilities of more than 60%. These drought episodes spread over almost the entire basin during the period 1971–2015 except in the central eastern areas (Daloa and Vavoua) and the western areas (Guiglo and Man) of the basin. Over this time period, the probability of obtaining two consecutive dry years is high, averaging 61.6%.

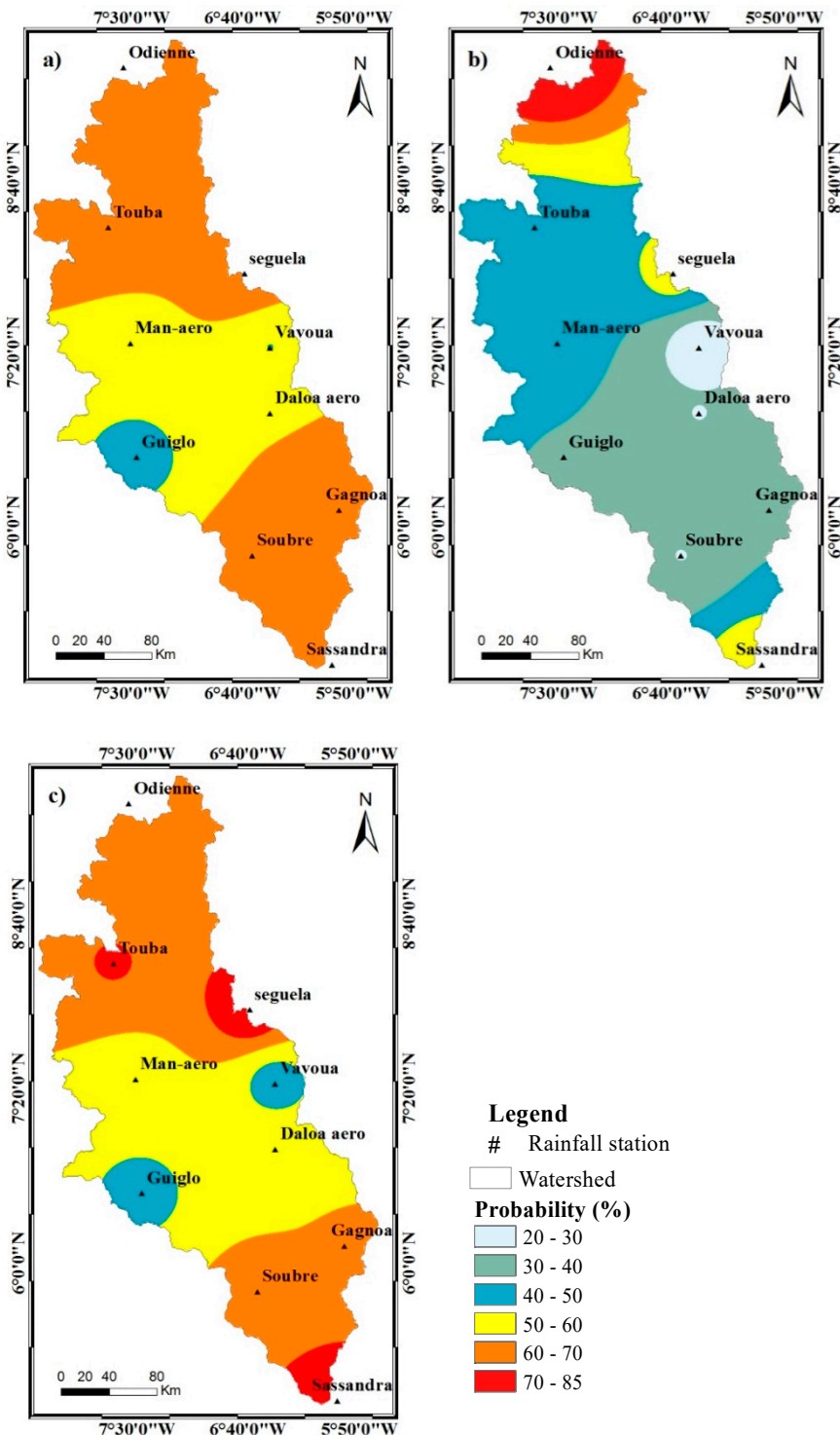

**Figure 3.** Spatial distribution of the occurrence for two successive dry years (D-D) over the Sassandra watershed (**a**) D-D probability over the period 1953–2015; (**b**) D-D probability over the period 1953–1970; (**c**) D-D probability over the period 1971–2015.

4.4.2. Spatial Variability of Probabilities for Three Consecutive Dry Years

The spatial variability of the probabilities of obtaining three successive dry years over the periods 1953–2015, 1953–1970 (before the rupture) and 1971–2015 (after the rupture) is presented in Figure 4. An intensification of droughts in the South (around the Sassandra, Gagnoa and Soubré regions) and

in the North-East (Seguéla) of the basin, with probabilities higher than 50%, is observed over the period 1953–2015.

Droughts intensify over time in the watershed. The dry areas which over the period 1953–1970 were located in the extreme North of the basin (covering the Odienné region), increased over the period 1971–2015 and spread with high probabilities of occurrence (up to 70%) over the Touba, Seguéla and South (Sassandra) regions. As for the northern tip of the basin, the probabilities were low.

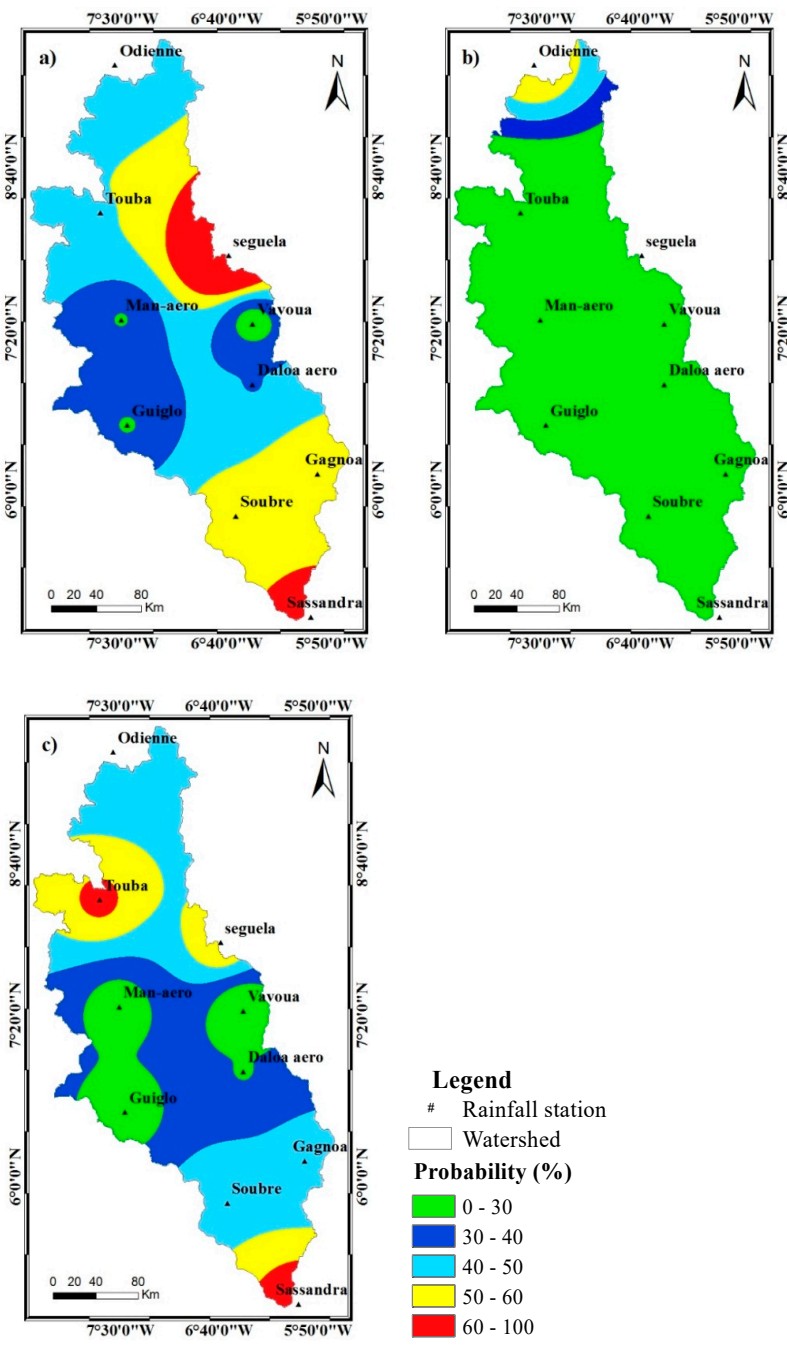

**Figure 4.** Spatial distribution of the occurrence for three successive dry years (D-D-D) over the Sassandra watershed (**a**) D-D-D probability over the period 1953–2015; (**b**) D-D-D probability over the period 1953–1970; (**c**) D-D-D over the period 1971–2015).

## 5. Discussion

The statistical test used to determine the breaks in the SPI series made it possible to detect breaks, for the most part, after the 1970s, which is the pivotal date observed almost throughout West Africa. The stations of Man, Guiglo and Gagnoa experienced early ruptures between 1967 and 1969. These years of disruption coincide with previous studies on rainfall in West Africa and Côte d'Ivoire [6,27,38–41]. This work reveals the appearance of a rainfall deficit from 1970 onwards and its continuation during the decades 1970–1979, 1980–1989 and 1990–1999. In the Sassandra basin, studies by [16] on the Lobo basin also mentioned a decrease in rainfall over the period 1970–2009 with a deficit of between 10% and 20% compared to the rupture years.

The analysis of the SPI index over the period 1953–2015 shows an increase of 54.35% to 82.35% in the number of dry years. This phenomenon was accentuated around the 1980s, with a persistence of dry sequences over the period 1970–2015.The peak of the most remarkable droughts occurred in 1974, 1983, 1987, 1987, 2002 and 2003. These peaks are characterized by "extremely severe" droughts. Moreover, changes in SPI values at the various stations indicate that, in terms of intensity, the regions of Man (−3.06), Guiglo (−2.86), Touba (−2.61), Vavoua (−2.26), Odienné (−2.05) and Soubré (−2.25), were the most affected by the droughts. As for duration, the SPIs indicate an increase in dry years after ruptures. Thus, Sassandra Station records the longest dry period (18 successive years). Our results are in line with studies conducted in West Africa. According to [42], the drought was more severe in the second half of the period 1900–2013, i.e., from the 1970s onwards. This trend is confirmed by several studies at the continental level and in West Africa [43,44]. These studies locate the most remarkable drought events in 1961, 1970, 1983, 1984, 1984, 1992 and 2001.

However, very few studies have focused on investigating the causes of droughts in West Africa. However, the work of [45,46] has shown that recent droughts remain linked to the emanation of ocean warming (southward warming gradient of the Atlantic Ocean and steady warming of the Indian Ocean) and fluctuations in the inter-tropical convergence zone (ITCZ). The ITCZ is defined as a convergence zone of northeastern Harmattan winds from the Sahara and the southwestern monsoon flow from the Atlantic [47]. The zenithal movement of the sun will command the displacement of the inter-tropical convergence zone towards the South to reach its southern position around 5° N in West Africa. This explains the introduction of the dry seasons. In addition to these factors, there is the effect of the land-atmosphere feedback through natural vegetation and land cover change. Indeed, deforestation can cause significant reduction of the rainfall and effect on the monsoon circulation [42]. The application of Markov chains over the periods 1953–2015 and the after change point detection year made it possible to highlight the areas that were most affected by droughts in terms of occurrence probability. The analysis shows that during the period 1953–2015 the probability of obtaining two successive dry years is high in the Northern and South of this basin. As for the probabilities of obtaining three dry years, they are high over part of the South and the northeast of the watershed. Over the periods 1953–1970 and 1971–2015, a large variation in probabilities was observed. Indeed, during 1953–1971, the dry areas that were observed in the extreme North of the basin according to Markov 1, spread over the entire basin over the period 1971–2015, with high values recorded in the North and South (ranging from 71% to 80%). As for the second order chains, they show the probabilities of obtaining high D-D-D in the far North over the period 1953–1970. During 1971–2015, drylands spread over part of the North (Touba), the North-East (Séguéla) and the South (Sassandra). These results indicate that the succession of dry conditions increased during the period 1971–2015 compared to the previous period (1953–1970) with the southern and northern areas of the basin most affected. This situation could be related to the effects of climate change observed on rainfall in West Africa. These results complement the work done of [35] on the transboundary watershed of the Bia River in eastern Côte d'Ivoire. The conclusions of this study are that the succession of two to three dry years is more marked in this basin after 1970.

The fact that the southern region located in the forest zone has the highest probability of two or three successive dry years is possibly due to the effect of droughts on the spatial and temporal variation of the 1200 isohyet in the southwestern part of the basin. According to [17], these variations

over the decades 1981–1990 and 1991–2000, created a dry corridor focused on the city of Sassandra, thus promoting a microclimate surrounded by watered areas that coincide with forest reserves and forest areas. In addition, there may be a significant reduction in forest cover in favor of an increase in cultivation areas, fallows and habitat.

## 6. Conclusions

Application of the standardized rainfall index (SPI) has made it possible to characterize drought situations in the Sassandra catchment area. During the 1971–2015 period, the driest sequences in terms of intensity (values between −1.53 and −3.06) and duration occurred. These dry events reached their peak in 1974, 1983, 1987, 1987, 2002 and 2003, with extremely severe droughts. Among the 10 stations that have been studied, those of Man, Guiglo, Touba, Odienné, Soubré, Seguéla and Vavoua seem to be the most affected by droughts. As for the Sassandra station, it seems to be more affected by a long dry period. As for the study of drought persistence using Markov chains 1 and 2, it made possible to determine the probability occurrence of droughts as well as to analyze their behavior in the basin. The results indicated that the succession of dry conditions increased during 1971–2015. The greatest probabilities for obtaining a doublet of successive dry years (D-D) and three consecutive dry years (D-D-D) were recorded in the southern and northern regions of the watershed. Markov chains 1 and 2 applied to a representative sample of 10 stations are found to provide a good regional drought indicator. Thus, the probability of a dry year in this basin will depend on the previous year's situation and even more on the condition of the year before. This study will enable populations, decision-makers, etc., to develop new water resource management strategies for the proper functioning of existing socio-economic infrastructures (hydroelectric and agricultural dams, etc.), water projects and the development of new farming systems to cope with the effects of climate change.

**Author Contributions:** N.S. and Y.A.N.G. developed the ideas; G.E.S. and N.D.H.M. contributed to the data processing. N.D.H.M. contributed to creation of the maps. N.S. analyzed the data and wrote the article with contributions from Y.A.N.G., N.E.S., N.D.H.M., and B.T.A.G.. Supervision and validation were provided by Y.A.N.G. and B.T.A.G.

**Funding:** This research received no external fundi.

**Acknowledgments:** The authors thank the National Meteorology Directorate (DMN) of SODEXAM (Company operating and Developing airports, Aeronautics and Meteorological) of Côte d'Ivoire, for data acquisition.

**Conflicts of Interest:** The authors declare no conflict of interest.

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
