# Peer review of "Characterization of Meteorological Droughts Occurrences in Côte d’Ivoire: Case of the Sassandra Watershed"

_climate, doi:10.3390/cli7040060_

Round 1

Reviewer 1 Report

General comments

The authors analyze the precipitation data of the Sassandra watershed (Cote d’Ivoire) in order to characterize the droughts sequences from 1953 to 2015. The work is of scientific interest and adequately introduced. However, there are some parts that need to be improved to increase the quality and the readability of the work. In particular the mathematical description is not clear and misplaced in the paper. Also, figures and tables must immediately follow the text when they are mentioned.

I suggest major revision.

All my comments are listed below.

Specific comments

Lines 39-42: This is a repetition of lines 36-39.

Line 46: “The example”, not “This example”.

Line 48: Delete the point after “areas”.

Line 48: [11-12] and the decrease…

Line 57: watershed not watershep.

Line 62: Man, with initial capital letter.

Line 64: Add a space before “The”.

Line 69: rainfall, not rainfull (please not that      this must be changed in many places in the manuscript).

Line 69: “The interannual rainfull average it      evaluate to 1441.5 mm” should be “The interannual rainfall average is      1441.5 mm”.

Line 89: Please explain what is a “break year”      from a physical point of view.

Line 92: Not clear what “square 1” means; it      refers to equation 1 towards the end of the manuscript. The authors have      placed all the equations in paragraph 3.3 without any descriptions. I      think it should be better to move the equations in paragraph 3.2 soon      after they are mentioned.

Line 97: Delete “(McKee et al., 1993)”, which is      already mentioned as [20].

Line 104: Remove “square 2” and all the “square      N” where they appear.

Line 128-129: The concept, with different words,      has already been written at lines 124-126.

Line 130: This paragraph, and the corresponding      sub paragraphs, must be numbered with 4, not 3.

Line 132: SPI must be written with all capital      letters.

Line 166: Add a point at the end of the sentence,      and remove the semicolon.

Line 193: Add a point after (D-D), and remove the      semicolon.

Line 197: “a trend contrary to the previous      period…”

Line 210: (D-D-D) does not appear in Table 8.

Lines 210-211: “Sassandra (67%) and Gagnoa      (59%)”.

Line 223: Add a point at the end of the sentence,      and remove the semicolon.

Lines 231-232: The sentence should be: “The      period 1953-2015 is marked by an increase in meteorological droughts over almost      the entire basin with the probability of having a very high doublet of dry      years that reaches 70%.”

Line 242: Use “break”, as in the rest of the      paper, in place of “rupture”. It appears twice in the sentence.

Line 251: In order to improve the readability of      the paper, it is better to put figures and tables soon after the text when      they are mentioned. Do not put all them in a specific paragraph.

Figure 2: The text “Break” must appear in all the      images.

Figure 3: Substitute “rainfull station” with      “rainfall station” or “precipitation station”.

Figure 3: Use the same probability levels in all      the three images. It will help the readers to compare the images.

Figure 4: Substitute “rainfull station” with      “rainfall station” or “precipitation station”.

Figure 4: Use the same probability levels in all      the three images. It will help the readers to compare the images.

Table 1: Use two digits everywhere: Gagnoa should      have a latitude of 6°07’. Please also add ‘.

Table 1: The last column to the right is not      useful because all the values are the same. Just add the information in      the table caption and/or in the text.

Table 2: Substitute “-0,99 à 0,99” with “-0.99 to      0.99”. Please use the point, not the comma, to indicate decimals. Same      observation for rows 2 and 3 of the table.

Table 3: Use “and”, not “at”.

Table 4: Use two digits for all the SPI values      (e.g. 1.90, not 1.9). Please use point not comma for decimals.

Table 7: “Climat soudanese” should be “Soudanese      climate”.

Table 7: “W-W-W” should be “D-D-D”.

Table 8: Substitute “Climat” with “Climate”      everywhere.

Table 8: “W-D-D” in the last column must be      “D-D-D”.

Table 8: Check the values 0,3 and 8,3 in column      D-D-W. All the other values are integers.

Line 360: The equations of paragraph 3.3 must be      moved in 3.2. They are not useful here.

 Equation      1: What is n and what is k? KS must be all capital. Use subscripts in the      text where they are needed.

Equation 2: (two lines after it): Where is S      used?

Equation 3: Use English words: “initial” not      “debut”.

Line 370: Use “standardized precipitation index”      in place of “rainfall standardized index” (in order to agree with the SPI      acronym).

Line 383: “after the break/change”.

Line 401: “… dry year is possibly…”

Lines 416-418: The final part of the sentence is      not clear: “…even more on the condition of the day before”. The probability      of a dry year depends on the condition of the day before?

Line 447: watershed, not watershep

Line 492: Spain with capital S.

Line 494: Hidden with capital H.

Line 500: If NWP stays for Numerical Weather      Prediction, it needs all capitals.

Author Response

Following the reviewers comments, the authors made a correction to the manuscript in order to improve it. The changes caused the rows to lines.

Point 1: Lines 39-42: This is a repetition of lines 36-39.

Response 1: Following the comments, the introduction was modified by authors. This sentence does not appear in this paragraph anymore.

Point 2: Line 46: “The example”, not “This example”.

Response 2:  this example was replaced by "the example" at line 56 following the modification of the manuscript.

Point 3: Line 48: Delete the point after “areas”.

Response 3: Punctuation was removed on line 58 after "areas".

Point 4: Line 48: [11-12] and the decrease…

Response 4: The sentence has been improved by adding and before the decrease… line 58.

Point 5: Line 57: watershed not watershep.

Response 5: "Watershep" was corrected by "watershed" in the manuscript, specifically at line 67

Point 6: Line 62: Man, with initial capital letter.

Response: Man was written with capital letter at the beginning, line 72.

Point 7: Line 64: Add a space before “The”.

Response 7: The space has been added after '' The '', line 74.

Point 8: Line 69: rainfall, not rainfull (please not that this must be changed in many places in the manuscript).

Response 8: Rainfull was rectified by rainfall in the manuscript

Point 9: Line 69: “The interannual rainfull average it evaluate to 1441.5 mm” should be “The interannual rainfall average is 1441.5 mm”.

Reponse 9: The sentence '' the interannual rainfull average it evaluate to 1441.5 '' has been replaced by '' interannual rainfall average it evaluate is 1441.5 mm ''. Line 81

Point 10: Line 89: Please explain what is a “break year” from a physical point of view.

Response 10: The year of rupture is the year when a significant change appeared in the series.

Point 11: Line 92: Not clear what “square 1” means; it refers to equation 1 towards the end of the manuscript. The authors have placed all the equations in paragraph 3.3 without any descriptions. I think it should be better to move the equations in paragraph 3.2 soon after they are mentioned.

Response 11: The word square has been replaced by equation. And the authors have inserted the equation into the text.

Point 12: Line 97: Delete “(McKee et al., 1993)”, which is already mentioned as [20].

Response 12: Mckee et al., 1993 was erased. Line 108.

Point 13: Line 104: Remove “square 2” and all the “square N” where they appear.

Response 13: All equations have been moved and inserted into the text.

Point 14: Line 128-129: The concept, with different words, has already been written at lines 124-126.

Response 14: Authors made a change in this part of the manuscript to avoid possible repetition.

Point 15: Line 130: This paragraph, and the corresponding sub paragraphs, must be numbered with 4, not 3.

Response 15: The numbering has been revised. From Line 193

Point 16: Line 132: SPI must be written with all capital letters.

Response 16: The standardized index of precipitation has been written all capital letters, SPI. Line 199.

Point 17: Line 166: Add a point at the end of the sentence, and remove the semicolon.

Response 17: Punctuation was added after the sentence. Line 237

Point 18: Line 193: Add a point after (D-D), and remove the semicolon.

Response 18: Punctuation was rectified after the sentence after (D-D). Line 269.

Point 19: Line 197: “a trend contrary to the previous period…”

Response 19:  Authors have corrected the sentence. Line 273.

Point 20: Line 210: (D-D-D) does not appear in Table 8.

Response 20: The authors have corrected the table. Table 8.

Point 21: Lines 210-211: “Sassandra (67%) and Gagnoa (59%)”.

Response 21: The sentence was improved by "Sassandra (67%) and Gagnoa (59%)". Line 294-295.

Point 22: Line 223: Add a point at the end of the sentence, and remove the semicolon.

Response 22: The semicolon was replaced by a point in the sentence. Line 316

Point 23: Lines 231-232: The sentence should be: “The period 1953-2015 is marked by an increase in meteorological droughts over almost the entire basin with the probability of having a very high doublet of dry years that reaches 70%.”

Response 23: The sentence has been improved. The authors have written, "The period 1953-2015 is marked by an increase in meteorological droughts over almost the entire basin with the probability of having a very high doublet of dry years that reaches 70%." Line 345-346.

Point 24: Line 242: Use “break”, as in the rest of the paper, in place of “rupture”. It appears twice in the sentence.

Response 24: The word rupture was replaced erased at line and replaced by '' change point detection '' in the manuscript.

Point 25: Line 251: In order to improve the readability of the paper, it is better to put figures and tables soon after the text when they are mentioned. Do not put all them in a specific paragraph.

Response 25: Figures, tables have been moved following paragraphs where they appear.

Point 26: Figure 2: The text “Break” must appear in all the images.

Response 26: Change point detection appears on all the images in Figure 2.

Point 27: Figure 3: Substitute “rainfull station” with “rainfall station” or “precipitation station”.

Response 27: Rainfull station has been replaced by '' rainfall station '' in Figure 3.

Point 28: Figure 3: Use the same probability levels in all the three images. It will help the readers to compare the images.

Response 28: Same legend was used for all the cards in Figure 3.

Point 29: Figure 4: Substitute “rainfull station” with “rainfall station” or “precipitation station”.

Response 29: Rainfull station was replaced by rainfall.

Point 30: Figure 4: Use the same probability levels in all the three images. It will help the readers to compare the images.

Response 30: The same legend was used for all the cards in Figure 4.

Point 31: Table 1: Use two digits everywhere: Gagnoa should have a latitude of 6°07’. Please also add.

Response 31: The coordinate’s values have been corrected in Table 1. Line 100.

Point 32: Table 1: The last column to the right is not useful because all the values are the same. Just add the information in the table caption and/or in the text.

Response 32: The last column of Table 1 has been deleted and the information has been introduced into the text. Line 100.

Point 33: Table 2: Substitute “-0,99 à 0,99” with “-0.99 to 0.99”. Please use the point, not the comma, to indicate decimals. Same observation for rows 2 and 3 of the table.

Response 33: In Tables 2 and 3 the points were used instead of commas.

Point 34: Table 3: Use “and”, not “at”.

Response 34: In Table 3 at was replaced by '' and ''.

Point 35: Table 4: Use two digits for all the SPI values (e.g. 1.90, not 1.9). Please use point not comma for decimals.

Response 35: The values in Table 4 have been corrected.

Point 36: Table 7: “Climat soudanese” should be “Soudanese climate”.

Response 36: “Climat Soudanese” was remplaced by “Soudanese climate.

Point 37: Table 7: “W-W-W” should be “D-D-D”.

Response 37: In table 7, W-W-W was remplaced by D-D-D.

Point 38: Table 8: Substitute “Climat” with “Climate” everywhere.

Response 38: Authors substituted  Climat with Climate.

Point 39: Table 8: “W-D-D” in the last column must be “D-D-D”.

Response 39: In table 8 D-D-W was substitute with D-D-D.

Point 40: Table 8: Check the values 0,3 and 8,3 in column D-D-W. All the other values are integers.

Response 40: In the DDW column, the values 0.3 and 8.3 have been corrected.

Point 41: Line 360: The equations of paragraph 3.3 must be moved in 3.2. They are not useful here.

Response 41: The equations have been moved following the paragraphs where they appear.

Point 42: Equation 1: What is n and what is k? KS must be all capital. Use subscripts in the text where they are needed.

Response 42: Following the different observations, Equation 1 was changed to Equation 3 in the manuscript. The test used has been replaced. From line 149.

Point 43: Equation 2: (two lines after it): Where is S used?

Response 43: In equation 3, S represents the standard deviation of the rainfall series over the period 1953-2015.

Point 44: Equation 3: Use English words: “initial” not “debut”.

Response 44: This observation has been taken into account in equation 2.

Point 45: Line 383: “after the break/change”.

Response 45: The authors substitute break/change with change point detection. Line 412.

Point 46: Line 401: “… dry year is possibly…”

Response 46: The sentence has been improved. Line 430.

Point 47: Lines 416-418: The final part of the sentence is not clear: “…even more on the condition of the day before”. The probability of a dry year depends on the condition of the day before?

Response 47: The phrase has been improved by replacing day by year. Line 445-447.

Point 48: Line 447: watershed, not watershep

Response 48: This observation has been taken into account in the manuscript. Watershep has been replaced by watershed. Line 486.

Point 49: Line 492: Spain with capital S.

Response 49: Spain was written with capital S. Line 531.

Point 50: Line 494: Hidden with capital H.

Response 50: Hidden was written with capital H. line 533.

Point 51: Line 500: If NWP stays for Numerical Weather Prediction, it needs all capitals.

Response 51: NWP has been written "Numerical Weather Prediction". Line 539.

Reviewer 2 Report

This is an interesting and important research to Cote d-Ivoire, that is an undeniable fact. Nevertheless, there are some appointments that the authors should observe:

1) Abstract and all manuscript need to have an intense grammar revision. I am not a native English but this proceeding is highly necessary;

- p. 03 (lines 88 to 92): confused paragraph. There is a need to be more didactic;

- p. 03 (lines 90 and 91): "This statistical processing will be performed with Hydrospect 2.0 software. The test statistic is defined as follows (square 1)". - what is square 1? Did you mean equation 1?

- What is the concept of break year or rupture period? What is Markov Chain (although it is a classical statistic method the authors should explain it better.)

- Why the mathematical proceedings are so far and so generalized on its explain? Why the mathematical sector is being located out of Data and Methods sector?

- I really suggest a reformulation of the Data and Methods sector. It is a confused sector which must be more organized in my opinion. Try to explain each step of the research and put the math equations together. Try to develop this sector with an unique body structure for the methodological explain from the beginning to the final part of the research.

- There is a Data and Methods sector and there are a Results sector with the same number (chapter three).

- Figure 1: it would be good a map with the location of Cote d'Ivoire in Africa. Moreover, the authors are using a geographical coordination type that is not a good one (7, 7.5, 8, 8.5 ...) at Sassandra Watershed. Replace it for a classical one (degrees, minutes, seconds ...).

- Figure 3: I think that the colors are not well chosen. Why not use the same pattern as figure 4 colors?

- P. 15 (lines 371-373): "The ITCZ is defined as a convergence zone of the northeasterly Harmattan winds that originate in the Sahara and the southwest monsoon flow that emanates from the Atlantic [41]". - I really think that the quality of the text is affecting the quality of this very important affirmation. Review this ITCZ information and review all text of the Discussionm sector as well.

- I strongly suggest a better explanation of the atmospherical characterization of Cote d´Ivoire (it could be insert at Study Area sector). This action would be very important to explain the ITCZ influence at Cote d´Ivoire because ITCZ dynamic is the answer and the key to comprehend the wet and dry distribution periods along the Sassandra watershed.

Author Response

Following the reviewers comments, the authors made a correction to the manuscript in order to improve it. The changes caused the rows to lines.

Point 1: Abstract and all manuscript need to have an intense grammar revision.

Response 1: The manuscript has been grammatically corrected.

Point 2: p. 03 (lines 88 to 92): confused paragraph. There is a need to be more didactic

Response 2: This paragraph has been revisited. The equations have been detailed. Line 140.

Point 3: p. 03 (lines 90 and 91): "This statistical processing will be performed with Hydrospect 2.0 software. The test statistic is defined as follows (square 1)". - What is square 1? Did you mean equation 1?

Response 3: Response 11: The word square has been replaced by equation. And the authors have inserted the equation into the text.  Line 148.

Point 4: What is the concept of break year or rupture period? What is Markov Chain (although it is a classical statistic method the authors should explain it better.)

Response 4: The year of rupture is the year when a significant change appeared in the series. A Markov chain is statistical test who is widely used for rainfall analysis and modelling.

Point 5: Why the mathematical proceedings are so far and so generalized on its explain? Why the mathematical sector is being located out of Data and Methods sector?

Response 5: the mathematical sector has been revisited and inserted after the paragraphs where they are mensioned.

Point 6: I really suggest a reformulation of the Data and Methods sector. It is a confused sector which must be more organized in my opinion. Try to explain each step of the research and put the math equations together. Try to develop this sector with an unique body structure for the methodological explain from the beginning to the final part of the research.

Response 6: Authors redrafted the data and methods sector. The point of the research are explained and detailed. The equations were inserted into the text.

Point 7: There is a Data and Methods sector and there are a Results sector with the same number (chapter three).

Response 7:  The numbering of the sector data and method and that of the results, has been corrected.

Point 8: Figure 1: it would be good a map with the location of Cote d'Ivoire in Africa. Moreover, the authors are using a geographical coordination type that is not a good one (7, 7.5, 8, 8.5 ...) at Sassandra Watershed. Replace it for a classical one (degrees, minutes, seconds ...).

Response 8: Figure of the Cote d'Ivoire localization has been redone using a geographical coordination in degree. The map of Cote d'Ivoire has also been located in Africa.

Point 9: Figure 3: I think that the colors are not well chosen. Why not use the same pattern as figure 4 colors?

Response 9: Same legend was used for all the cards in Figure 3. Rainfull station was replaced by rainfall. The authors chose different legends to differentiate D-D probabilities (figure 3) from D-D-D probabilities (figure 4).

Point 10: I strongly suggest a better explanation of the atmospherical characterization of Cote d´Ivoire (it could be insert at Study Area sector). This action would be very important to explain the ITCZ influence at Cote d´Ivoire because ITCZ dynamic is the answer and the key to comprehend the wet and dry distribution periods along the Sassandra watershed.

Response 10: ITCZ information on the Cote d'Ivoire has been included in the area of the study area. Line 75.

Reviewer 3 Report

The presented manuscript deals with the important issue of proposing mitigation or adaptation measures for water management. The form of the manuscript is not article, it is rather case report or typical ase of study of occurrences meteorological droughts in Côte d’Ivoire: Case of the Sassandra watershed. Article is poor prepared, almost all figures are illegible, for example: Line 275. Figure 1. Geographic localization of the study area and line 301. Figure 3. Spatial distribution of the occurrence for two successive dry years (D-D) over the Sassandra catchment area (a: D-D probability over the period 1953-2015; b: D-D probability over the period 1953-1970 (before break year); c: D-D probability over the period 1971-2015 (after break year)). Also the introduction should include information about problem, which should be investigated, as well as reasons for conducting the research. The purpose of the work should be write more clearly. The manuscript should be prepared according to journal guidelines, and should be really carefully edited – why all tables and figures are pasted in the separeted point? I did not find such guideline in the section of the Instructions for Authors, and it is difficult to follow, when You did not see the figures directly the text.The introduction and the conclusion should include the value added with respect to existing research. Some statistics of the presented results should be included in the discussion of results, concerning for example the SPI index evolution over the period 1953-2015. Line 342. In the table some french words occur, it should be presented in English. According to the typographic rules, the decimal part is separated by a dot, and the comma is used, for example, to separate thousands, so the whole text, figures, tables should be prepared according to this rule. Line 360, Authors can’t start the section from the equation, some introduction of this equation should be firstly presented. On what base You assumed that Markov chains will be useful to determine the occurrence probabilities of successive dry years? All equations in the section 3.3. Formatting of Mathematical Components, should have the same font as the text. Eq. 3. Translate debut in English, etc, and the rest of the French expressions. Calculations of the conditional probabilities should be presented after Eq. 7. How You calculate the conditional probabilites, some errors occur, a sto check it send the .xls file for the whole performed calculations. Some information about practical use of the obtained results should be presented, both in the section Discussion and Conclusions should be underlined.

Author Response

Following reviewer, a modification of the manuscript was made. Introduction, methodology, results, and discussion have been corrected. Thus:

Point 1: Line 275. Figure 1. Geographic localization of the study area and line 301.

Response 1: This card has been corrected to facilitate reading. Area study sector (line 70).

Point 2: Also the introduction should include information about problem, which should be investigated, as well as reasons for conducting the research. The purpose of the work should be write more clearly.

Response 2:  The authors made changes in the introduction and discussion.

Point 3: The manuscript should be prepared according to journal guidelines, and should be really carefully edited – why all tables and figures are pasted in the separeted point? I did not find such guideline in the section of the Instructions for Authors, and it is difficult to follow, when You did not see the figures directly the text.

Response 3: The manuscript has been grammatically corrected. This paragraph has been revisited. The equations have been detailed (data and methods, sector) Line 92.The word square has been replaced by equation. The mathematical and figures sector, has been revisited and inserted after the paragraphs where they are mentioned. The equations have been moved following the paragraphs where they appear.

Point 4: Some statistics of the presented results should be included in the discussion of results, concerning for example the SPI index evolution over the period 1953-2015.

Response 4: The discussion sector has been changed. Some statistics have been included in the discussion of results.

Point 5: Line 342. In the table some french words occur, it should be presented in English. According to the typographic rules, the decimal part is separated by a dot, and the comma is used, for example, to separate thousands, so the whole text, figures, tables should be prepared according to this rule.

Response 5: The values in Table 2 have been corrected. The points (.) were used to replace the commas. Line 125.

Point 6: Line 360, Authors can’t start the section from the equation, some introduction of this equation should be firstly presented.

Response 6: Introduction of this equation has been presented firstly in data and methods sector. Line 92.

Point 7: On what base you assumed that Markov chains will be useful to determine the occurrence probabilities of successive dry years? All equations in the section 3.3. Formatting of Mathematical Components, should have the same font as the text.

Response 7:  The Markov chain method was used in this study. This method is commonly used in the world according to some authors. It makes it possible to follow the persistence of droughts. In Ivory Coast very few studies have been done with this method.

Point 8: Equation 3. Translate debut in English, etc, and the rest of the French expressions.

Response 8:  The expressions in this equation have been corrected. Début and fin have been translated English. Line 152-192

Point 9: Calculations of the conditional probabilities should be presented after Eq. 7. How you calculate the conditional probabilites, some errors occur, a sto check it send the .xls file for the whole performed calculations.

Response 9: Conditional probabilities were calculated based on the basis the detailed mathematical equation in the data and methods, sector. These calculations were performed on Excel file.

Round 2

Reviewer 1 Report

General comments

The authors have reviewed the first version of the manuscript, but I still have some concerns about the form. For example, in some parts of the paper French is used in place of English, some sentences are not easily readable, another is duplicated.

My comments are listed below.

Specific comments

Line 21: Review the sentence (the by).

Line 47: Delete “etc…”

Lines 47-49: Please review the whole      sentence from “For example” to “[13]”. My impression is that the sentence      is not easily readable since some parts are missing and others are in      excess. For example: “… crops were destroyed”, “… people were affected”.      Is “burned” needed after “plantations”?

Line 49: The comma for separating thousands      must always be used (as in 15,000) or never (as in 3000).

Lines 52-55: The sentence has already been      written in lines 49-52.

Line 62: “and are” or “that are”?

Lines 73-74: As observed before, the comma      for separating thousands must always be used (as in 67,000) or never (as      in 75000).

Figure 1: The legend is in French.

Line 98: Review the sentence: “a good      homogeneous coverage” or “the most homogeneous coverage”?

Line 118: “over year i”, delete “one”.

Line 129: Duration is an important …

Line 134: Review the sentence. You used Ainitial      in place of Aend, and Adebut in place of Ainitial.      Also, remember to use English terms.

Lines 143-144: Is “Cumulative Gap” CS or      CG?

Line 147: Replace “will be” with “is”.

Line 149: Remove the French terms from the      equation and replace them with English ones.

Lines 150-151: Please define “n”.

Line 160: English language must be used.

Equation 4: Please use “|” in place of “/”.

Equation 5: Remove the French terms from      the equation and replace them with English ones.

Line 182: English language must be used.

Equation 8: Please use “|” in place of “/”.

Line 200: The sentence should end with “…      years (Figure 2).”

Line 225: Is “Thus:” needed?

Line 265: Table 6.

Line 318: “this probability (D-D-D)” should      be “the probability to observe D-D-D”.

Author Response

Response to Reviewer 1 Comments

Following the reviewers comments, the authors made a correction to the manuscript in order to improve it.

Point 1: Line 21: Review the sentence (the by).
Response 1: Sentence has been corrected (line 21).

Point 2: Line 47: Delete “etc…”
Response 2: This word has been deleted (line 47).

Point 3: Lines 47-49: Please review the whole sentence from “For example” to “[13]”. My impression is that the sentence is not easily readable since some parts are missing and others are in excess. For example: “… crops were destroyed”, “… people were affected”. Is “burned” needed after “plantations”?
Response 3: This sentence has been corrected, excessive expressions have been deleted.

Point 4: Line 49: The comma for separating thousands must always be used (as in 15,000) or never (as in 3000).

Response 4: The punctuation points were used instead of commas.

Point 5: Lines 52-55: The sentence has already been written in lines 49-52.
Response 5: The repeated sentence has been deleted. Line 53-56.

Point 6: Line 62: “and are” or “that are”?
Response 6: Sentence has been improved to make it more comprehensible to the reviewers. It is the Sassandra basin that is subject to many water-related projects (line 63).

Point 7: Lines 73-74: As observed before, the comma for separating thousands must always be used (as in 67,000) or never (as in 75000).
Response 7: The punctuation points were used instead of commas (line 74-75).

Point 8: Figure 1: The legend is in French
Response 8: the legend has been translated into French (line 92).

Point 9: Line 98: Review the sentence: “a good homogeneous coverage” or “the most homogeneous coverage”?
Reponse 9: the sentence has been reviewed. The authors have written: These … selected to provide the most homogeneous coverage of the different climatic areas across in the Sassandra basin (line 99).

Point 10: Line 118: “over year i”, delete “one”.
Response 10: the word “one” has been delete (line 117).

Point 11: Line 129: Duration is an important
Response 11: the word “an” has been added after duration is (line 128).

Point 12: Line 134: Review the sentence. You used Ainitial in place of Aend, and Adebut in place of Ainitial. Also, remember to use English terms.
Response 12: The term in French have been translated into English (line 133).

Point 13: Lines 143-144: Is “Cumulative Gap” CS or CG?
Response 13: It is CG (Cumulative Gap) (line 144).

Point 14: Line 147: Replace “will be” with “is”.
Response 14: “will be” have been replaced with “is”

Point 15: Line 149: Remove the French terms from the equation and replace them with English ones.
Response 15: The term in French have been translated into English (line 149).

Point 16: Lines 150-151: Please define “n”.
Response 16: “n” have been define: n is the number of value for the rank I (line 151).

Point 17: Line 160: English language must be used.
Response 17: English language have been used (line 169).

Point 18: Equation 4: Please use “|” in place of “/”.
Response 18: The equation has been corrected (line 163-169).

Point 19: Equation 5: Remove the French terms from the equation and replace them with English ones.
Response 19: English language have been used (line 169).

Point 20: Line 182: English language must be used.
Response 20: English language have been used (line 182).

Point 21: Equation 8: Please use “|” in place of “/”.
Response 21: The equation has been corrected (line 185).

Point 22: Line 200: The sentence should end with “… years (Figure 2).”
Response 22: SPI values over the 1953-2015 period show very few dry sequences before the rupture years (figure 2)… line 200.

Point 23: Line 225: Is “Thus:” needed?
Response 23: “thus” has been deleted at the end of the sentence (line 225).

Point 24: Line 265: Table 6.
Response: “in” was deleted before “table 6” (line 265).

Point 25: Line 318: “this probability (D-D-D)” should be “the probability to observe D-D-D”.
Response 25: “this probability (D-D-D)” has been remplaced with “the probability to observe D-D-D”.

Reviewer 2 Report

In this second version, I could observe some improvements at the math sector (it is better organized than last version). Nevertheless, the authors do not dared a better ITCZ synoptic analysis to understand why the reason of these SPI results within the historical context. In other words, this paper has a pure statistical aspect (ok, I understand).

Only two warnings:

P. 04, line 133 - I think that there is a mistake at equation 2 at the elements description. Check it.
P. 05, line 160; 182 - subtitles that are written in French.Translate to English, please.

There is no doubt about the importance of this research to Cote d´Ivoire, to Africa and to climatological and geographical studies as well.

Author Response

Response to Reviewer 2 Comments

Following the reviewers comments, the authors made a correction to the manuscript in order to improve it.

Point 1: P. 04, line 133 - I think that there is a mistake at equation 2 at the elements description. Check it.
Response 1: The descriptive term in French have been translated into English (line 133).

Point 2: P. 05, line 160; 182 - subtitles that are written in French. Translate to English, please
Response 2: English language have been used (line 160 and line 182).

Point 3: the authors do not dared a better ITCZ synoptic analysis to understand why the reason of these SPI results within the historical context.
Response 3: In a discussion section, editors briefly explained the impact of ITCZ on dry season implementation. Detailed analysis on the impact of ITCZ fluctuations requires further study. This study examines the characterization of meteorological droughts occurrence o. It focuses more on the statistical aspect of drought

Reviewer 3 Report

Some french words occur, it should be presented in English, as in line 160: Chaîne de Markov à deux états d’ordre. 
Are the distinguished
events independent? How many states are in this system and how are the transitions between states defined? Send the .xls file for the whole performed calculations.

Author Response

Response to Reviewer 3 Comments

Following the reviewers comments, the authors made a correction to the manuscript in order to improve it.

Point 1: Some french words occur, it should be presented in English, as in line 160: Chaîne de Markov à deux états d’ordre
Response 1: the words or sentences in French, have been translated into English in the manuscript (line 160 and line 182).

Point 2: Are the distinguished events independent?
Response 2: For a Markov string of order 2, the state of the variable E (t) at time t depends on its state E (t-1) at time (t-1) as well as its state E(t-2). The status of a year depends on the status of the previous year. But at the different stations, the drought phenomenon is independent. For example, the probabilities of having successive dry years in Sassandra are higher than those of Vavoua or Daloa.

Point 3: How many states are in this system and how are the transitions between states defined?
Response 3: for the Markov chain 1, we have four states (P00, P01, P10, and P11). For Markov chain 2, we have 8 states (P000, P010, P001, P100, P110, P111, P101 and P011), but the authors have presented in this manuscript the most significant states in terms of probability (table 3). Values 0 and 1 represent the dry year and the wet year respectively. For example, P000 represents the probability of obtaining three successive dry years (D-D-D).

Point 4: Send the .xls file for the whole performed calculations.
Response 4: The calculations were made on the basis of the formulas detailed in the manuscript. Like the SPIs, this was done on the basis of an algorithm implemented
